# Brief Communication: Mesoscale and submesoscale dynamics in the marginal ice zone from sequential SAR observations

Igor E. Kozlov[1], Evgeny V. Plotnikov[1], Georgy E. Manucharyan[2]

[1]Remote Sensing Department, Marine Hydrophysical Institute of RAS, Sevastopol, 299011, Russia
[2]School of Oceanography, University of Washington, Seattle, 98195-7940, United States of America

*Correspondence to*: Igor E. Kozlov (igor.eko@gmail.com)

**Abstract.** New possibilities for horizontal current retrieval in marginal ice zones (MIZs) from sequential Sentinel-1 synthetic aperture radar (SAR) images are demonstrated. Daily overlapping SAR acquisitions within 70-85° S/N at time intervals <1 hour enable estimation of high-resolution velocity fields, revealing MIZ dynamics down to submesoscales. An example taken from the Fram Strait MIZ reveals energetic eddies and filaments with Rossby numbers reaching O(1) magnitudes. The SAR-derived velocity estimations at such high spatial resolution can be critical for monitoring the evolving MIZ dynamics and model validation of submesoscale processes in polar oceans.

## 1 Introduction

Marginal ice zones (MIZs) are regions of strong lateral buoyancy gradients, energetic atmosphere-ice-ocean interactions and enhanced biological productivity (e.g. von Appen et al., 2018). With continuing global warming, MIZs remain a major source of uncertainties in sea ice prediction models (Tietsche et al., 2014) and have a strong widening trend observed in summer during the past several decades in the Arctic Ocean (Strong and Rigor, 2013). MIZ dynamics and thermodynamics are critically affected by mesoscale and submesoscale eddies that are formed due to a multitude of possible atmosphere-ice-ocean interaction mechanisms (e.g. Johannessen et al., 1987). Historical field campaigns have documented the presence of mesoscale eddy-related motions down to several hundred meters depth with associated mean orbital speeds of about 0.5 m s$^{-1}$ in the Fram Strait MIZ (Wadhams and Squire, 1983; Johannessen et al., 1987). These ice-edge eddies were observed to mechanically sweep the ice away from the ice pack and to entrain the warm Atlantic Water (AW) beneath the ice, causing an average ice edge retreat at a rate of 1-2 km day$^{-1}$ in the summer (Johannessen et al., 1987).

More recent model and field experiments also indicate energetic motions associated with submesoscale eddies, fronts and filaments along the Arctic and Southern Ocean MIZs (Manucharyan and Thompson, 2017; von Appen et al., 2018; Swart et al., 2020), which importantly impact the heat and carbon exchanges between the ocean and atmosphere. In particular, submesoscale ocean flows induce large vertical velocities that can entrain nutrients and relatively warm subsurface waters into the mixed layer with net vertical heat fluxes toward the ice surface reaching 100 W m$^{-2}$ (Manucharyan and Thompson,

2017). They also lead to enhanced mixing of water masses over short horizontal scales, thus, impacting the sea ice and biology structures within the MIZ (von Appen et al., 2018).

Surface signatures of MIZ eddies and filaments can be effectively traced in high-resolution synthetic aperture radar (SAR) images due to characteristic patterns formed by sea ice that mimics the ocean current structure beneath (Shuchman et al., 1987; Kozlov et al., 2019). The manifestation of MIZ dynamics in SAR images is caused by the different level of radar backscatter from open water and low-concentration sea ice that tend to accumulate along the surface current convergence zones at eddy boundaries. Unlike the surfactant films that outline the eddy boundaries primarily under low winds (e.g. Karimova and Gade, 2016), SAR signatures of sea ice are well sustained even under high wind conditions (Johannessen et al., 1987), providing almost all-weather capability for monitoring MIZ dynamics. In regions with relatively low ice concentrations (<50%), the sea ice tends to accumulate in predominantly cyclonic eddies and filaments, and the sea ice velocity becomes close to that of the surface ocean currents (Manucharyan and Thompson 2017), allowing one to make inferences about the upper ocean eddy dynamics from satellite sea ice observations.

Attempts to use spaceborne SAR data to retrieve sea ice motion in the polar oceans have been made since the launch of Seasat in 1978 (e.g. Hall and Rothrock, 1981), but relatively long sensing intervals (> 3 days) allowed to retrieve sea ice motions at relatively large scales, O(50-100 km). Having optimum capabilities to observe polar regions independent of illumination and weather conditions, the high-resolution SAR measurements are capable to resolve MIZ dynamics at significantly smaller scales, O(0.1-1 km), provided the time delay between sequential SAR images is sufficiently small, e.g. within 1-2 hours. However, until recently, such a combination was not realized in practice for non-commercial SAR missions.

The aim of this letter is to demonstrate a new possibility for regular SAR observations over the large MIZ regions that has recently become available from sequential measurements of the European Radar Observatory for the Copernicus joint initiative of the European Commission (EC) and the European Space Agency (ESA) Sentinel-1 A and B SAR missions launched in 2014 and 2016, respectively. As will be shown below, these data allows retrieving the high-resolution sea ice velocity field to observe the MIZ dynamics down to submesoscales on a daily basis. We believe that this information is critical for better understanding of the key dynamical processes governing the submesoscale variability in MIZs, as well as for improving and validation of sea ice and coupled ice-ocean models.

## 2 Data and methods

Since the launch of the Sentinel-1B satellite in 2016, the SAR data from the two European polar-orbit Sentinel-1A and Sentinel-1B missions have become available for the public access. Each Sentinel-1 satellite carries a C-band SAR instrument operating at multiple sensing modes, each having a certain spatial resolution, a range of incidence angles, and a set of polarization channels.

Due to their polar orbits, Sentinel-1 A and B have high measurement frequency over the high-latitude regions. Many of the Sentinel-1 images overlap within the latitude band 70-85° in the southern and northern hemispheres, forming a distinct set of sequential SAR observations with a time lag of just around 50 minutes. As a result, 2 to 4 overlapping scenes can be available on a daily basis over certain regions, such as the European Arctic Ocean (Fig. 1, a). Fig. 1 (a) shows a map of the western Eurasian Arctic with 43 Sentinel-1 A/B acquisitions available on 17 September 2017 at the Copernicus Open Access

Hub (https://scihub.copernicus.eu). As is clearly seen, significant portions of the region, increasing northward, are covered by overlapping SAR scenes. The relatively short time lag between consecutive measurements, O(1 h), and the high spatial resolution of SAR data, O(100 m), provide  a unique opportunity to observe the MIZ dynamics on the daily basis.

We analyze Sentinel-1 SAR images acquired in September 2017 over Fram Strait (FS) (green frame in Fig. 1, a), the region of quasi-permanent ice edge formed between the warm Atlantic Water (AW) flowing northward into the Arctic Ocean and

cold ice-covered Polar Water (PW) flowing southward (von Appen et al., 2016). It should be noted that our study period corresponds to the end of the melt season during which the formation of a shallow mixed layer in the MIZ might favor the presence of various mesoscale and submesoscale dynamic features. Though there might be a certain seasonal variability of such dynamic features seen in SAR data (e.g. Kozlov et al., 2019), this is not addressed here. The Copernicus Hub shows 120 SAR images available over FS in September 2017, about half of which are forming pairs of partly overlapping

sequential images. Here we focus on a pair of Sentinel-1 images acquired on September 17, 2017 at 07:12 UTC (Sentinel-1A) and at 8:00 UTC (Sentinel-1B) with a time lag of 48 minutes (Fig. 1, b). The data are gridded Level 1 Extra-Wide swath mode medium-resolution (~90 m) products covering an area of ~400×400 km at HH and HV polarisations. The dual-polarized HH-band is further used for processing and analysis.

To estimate the velocity field from a pair of sequential SAR images, we implement the following procedure: i) image

calibration for every image in the pair, ii) selection of overlapping image fragments, their normalization and filtering, iii) calculation of horizontal velocity field for image fragments using one of the methods for velocity estimation from image sequences (e.g. Emery et al., 1986; Chen, 2011; Marmorino and Chen, 2019). The Sentinel-1 images are calibrated to obtain the normalized radar cross-section units. The overlapping fragments of both images in the pair are normalized to remove the signal trend in the range direction, and finally smoothed to reduce the speckle noise using the adaptive Wiener filter. The

major issue arising at this step is the change in the SAR viewing geometry between sequential images. As a result, the level of radar backscatter over the particular surface area would differ from image to image, and so would affect the clarity of eddy manifestation in SAR images. This issue is addressed during the normalization step, but might be difficult to overcome for very thin ice whose backscatter is very sensitive to the described changes in the viewing geometry and near-surface winds. For demonstration, here we use the maximum cross-correlation method (MCC) (Emery et al., 1986; Qazi et al., 2014)

to retrieve the surface velocity vectors, but we acknowledge that more elaborated velocimetry methods can also be used for this purpose (see e.g. Chen, 2011; Marmorino and Chen, 2019). The preliminary analysis of SAR data from various dates in summer 2017 suggests that MCC works rather effectively for typical sea ice concentrations encountered along the ice edge and in the marginal ice zone, provided the movement of ice floes is apparent in the sequential SAR images. The MCC

method was used with a moving window from 25×25 pixels for the initial large overlaps and down to 3×3 pixels for zooms over small-scale features with maximum allowed shifts up to 25 pixels in the zonal and meridional directions. For the pixel size of 40 m, the resulting velocity fields were obtained at 1 km and ~100 m resolution, respectively. Given the spatial resolution of the SAR images of 88×87 m in range and azimuth directions, pixel spacing of 40 m, and the time lag between sequential images equal to 48 minutes, the velocity detection threshold in this case would be 0.03 m s$^{-1}$ ± 0.01 m s$^{-1}$, similar to (Marmorino and Chen, 2019).

## 3 Results

### 3.1 Structure of MIZ

The positions of the two sequential Sentinel-1 SAR images acquired on September 17, 2017 over Fram Strait are shown in Fig. 1 (b). The area of the FS MIZ estimated directly from the SAR images is equal to ~15 000 km$^2$ with an average width of 60-70 km. The region contains a large number of small-to-mesoscale eddies, filaments and meanders, with some eddies appearing in the form of cyclone-anticyclone dipoles that are expected to form via instability of outcropping fronts (Manucharyan and Timmermans, 2013). Figure 1 (c) shows an enlarged fragment of the Sentinel-1A image that clearly shows the manifestation of an anomalously large anticyclonic vortex (marked as **A** in Fig. 1, c) with a diameter of about 80-90 km spreading southward out of the main MIZ over the depths of around 2000 m. One can clearly see the formation of another smaller cyclonic vortex **C1** with a diameter of about 15 km on the western periphery of **A**. Many other small-scale eddies and meanders O(1 km) are seen along its periphery (Fig. 1, c). Notably, the periphery of the anticyclone **A** is bounded by several curved ice-filled narrow filaments, while a lot of open water with dispersed low-concentration ice fields is found in the center. This is opposite for the cyclonic eddies **C1** and **C2** with more ice accumulated in their centers, in agreement with modeling studies demonstrating the preferential sea ice accumulation in cyclones (Manucharyan and Thompson, 2017).

### 3.2 Velocity retrieval over MIZ

A fragment of the SAR image presented in Fig. 1 (c) was further used for surface current estimation using the MCC method. According to the WindSat and the ASCAT scatterometer data for 17 September 2017 (not shown), the wind conditions were characterized by low south-easterly winds of 3-5 m s$^{-1}$, under which the ice drift near the ice edge should reflect the underlying ocean circulation (Shuchman et al., 1987; Manucharyan and Thompson, 2017).

Fig. 2 (a) shows the resulting velocity field obtained for the initial image fragment with the large anticyclonic vortex **A**. The overall geometry of the obtained current field is in a good agreement with the ice structures seen in the SAR image, and shows a pronounced anticyclonic rotation associated with the large vortex **A**. The modulus of horizontal current velocity $|\boldsymbol{u}|$, comprised of the eastward $u$ and the northward $v$ velocity components, is shown in Figure 2 (b). As seen, a general southeastward drift of the MIZ with an average velocity of 0.2-0.3 m s$^{-1}$ is seen in the upper part of the image. It increases in the middle of the scene and forms a strong southward jet with current velocities reaching 0.75 m s$^{-1}$. This jet-like structure

then evolves into the large anticyclonic eddy **A** downstream. The mean orbital velocity of vortex **A** is about 0.4-0.5 m s$^{-1}$. However, the maximum values attain 0.65-0.75 m s$^{-1}$ along its north-western periphery and 0.5 m s$^{-1}$ along its southern boundary, gradually decreasing toward the center (Fig. 2, b). Such high velocity values are confirmed by the manual analysis of horizontal shifts of individual ice floes in sequential images (not shown).

Fig. 3 (a) shows the velocity field for the enlarged SAR fragment over the western part of the anticyclone **A**. A number of distinct dynamic features are seen along its periphery, including the cyclonic vortex **C1** and the narrow elongated filaments **F1** and **F2**. As seen from Fig. 3 (a), all these features are well manifested in the SAR image due to enhanced radar backscatter from sea ice. The sea ice in the narrow filaments is likely accumulated due to surface current convergence zones associated with ageostrophic secondary circulation near submesoscale fronts and filaments (McWilliams, 2016). The field of horizontal divergence, $\nabla \cdot \boldsymbol{u} = \partial u/\partial x + \partial v/\partial y$, confirms the formation of strong convergence zones up to -5×10$^{-4}$ s$^{-1}$ (blue color in Fig. 3, b) that correspond to bright sea ice patterns seen in the SAR image (Fig. 3, a). Thus, the presence of filamentary sea ice patterns is associated with regions of strong surface convergence and downwelling.

As noted above, the periphery of the anticyclone **A** is bounded by several narrow filaments (Fig. 3, a). Filament **F1** is 0.5-1.5 km wide and ~60 km long, very similar to the submesoscale cyclonic filament sampled in detail in the Fram Strait MIZ by von Appen et al. (2018) where velocities of $\pm$0.5 m s$^{-1}$ were observed with a vessel mounted acoustic Doppler current profiler. However, as both the interpretation of the SAR image and the retrieved current velocity suggest, this filament is not a stand-alone feature, but is a part of the larger eddy-induced frontogenesis pattern and, hence, cannot be interpreted out of the context. The important consequence is that its propagation direction and, hence, the sign of relative vorticity, $\zeta = \partial v/\partial x - \partial u/\partial y$, is different depending on its part to be considered (Fig. 3, c).

The data show that **F1** is stretching along-front in opposite directions and moves northward with a mean (maximum) speed of 0.4-0.5 (0.75) m s$^{-1}$ in its upper half, while drifting southward at an average (maximum) velocity of 0.3-0.4 (0.55) m s$^{-1}$ in the lower part, as shown by arrows in Fig. 3 (a). The cross-front velocity is almost negligible in its southern and northern parts, while it attains ~0.05-0.1 m/s near the divergence point (found next to **F1** notion in Fig. 3, a). The relative vorticity $\zeta$ values estimated for the mean current velocity $\Delta v = 0.4$ m s$^{-1}$ and characteristic filament cross-front width $\Delta x = 1$ km are $\zeta = \Delta v/\Delta x = \sim 3f$, where $f$ is the Coriolis parameter equal $f = 1.433 \cdot 10^{-4} s^{-1}$ for latitude $\theta = 80.2°$ N. This gives the Rossby number $Ro = \zeta/f \sim O(1)$, clearly indicating the submesoscale nature of this filament. Similar stretching and movement into the opposite directions are observed for the filament **F2**, which also splits into the cyclonic and anticyclonic counterparts (Fig. 3, c). Its cyclonic part starts to meander and then rotates anticlockwise to shape the boundary of the cyclone **C1**.

The shape of the cyclone **C1**, defined from the orientation of the bounding ice streaks, is highly elliptical with the minor and major axis being about 10 km and 25 km, respectively (Fig. 3, a). The associated orbital velocities are 0.2-0.65 m s$^{-1}$ (mean value 0.5 m s$^{-1}$), being largest along its major axis on the western and eastern sides, and smallest near the eddy center. The horizontal divergence field shows high negative values (convergence) where sea ice accumulates along the eddy boundaries (compare Figs. 3a and 3b). As noted above, the ice concentration at the boundaries and in the center of **C1** is higher than for

eddy **A**. This is confirmed by the more intense surface convergence over **C1** (Fig. 3, b) that is presumably linked to stronger ageostrophic motions. Indeed, the comparison of vorticity values for eddies **A** ($\zeta \approx 0.07f$) and **C1** ($\zeta \approx 0.3f$) yields a larger Rossby number than for the cyclone **C1** with ageostrophic effects playing a higher role in this case.

Fig. 3 (d) shows the field of instantaneous kinetic energy (KE), $KE = \frac{1}{2}(u^2 + v^2)$. In general, one sees very energetic patterns associated with eddy dynamics in the MIZ with the mean KE of about 0.1 m² s⁻² and the maximum of 0.23 m² s⁻² over the northern periphery of **A**. For the cyclone **C1**, the KE is slightly less, but still high: the maximum value of 0.2 m² s⁻² is found over the western periphery of **C1**, while over the eastern part it is about 0.1-0.15 m² s⁻².

## 4 Discussion and conclusions

We demonstrated a new possibility to retrieve horizontal velocity fields from sequential Sentinel-1 SAR images taken over low-concentration ice regions where sea ice motion is indicative of mesoscale and submesoscale eddies and filaments. The pair of Sentinel-1 SAR images acquired over the Fram Strait MIZ revealed a large anticyclonic ice-edge eddy of 80-90 km in diameter and numerous cyclonic eddies of smaller size at its periphery, bounded by several elongated ice-filled filaments. The reconstructed currents revealed strong convergence zones and relative vorticity magnitudes corresponding to O(1) Rossby numbers. While historic field campaigns have documented the generation of ice-edge eddies in central Fram Strait (e.g. Wadhams and Squire, 1983; Johannessen et al., 1987; Shuchman et al., 1987), the observations of such an anomalously large and energetic MIZ eddy in the Arctic Ocean have never been reported before.

Notably, the location of MIZ eddies reported here coincides with the ice edge region in central FS characterized by high summer-time eddy kinetic energy (EKE) values (exceeding 0.02 m² s⁻²) reported by Bulczak et al. (2014) based on satellite altimetry data, and later confirmed by long-term mooring observations (von Appen et al., 2016). Such anomalously high EKE values were attributed to the complex atmosphere-ice-ocean interplay, including the formation of eddies due to barotropic and baroclinic instability of an ice edge jet along the MIZ, topographic generation and trapping, interaction of AW eddies advected to the ice edge with meltwater fronts, wind-induced differential Ekman pumping along a meandering ice edge, or their combinations (Johannessen et al., 1987). The lifetime of such eddies was reported to be at least 20-30 days with diameters ranging within 20-40 km, rarely reaching 60 km (Wadhams and Squire, 1983). In our case, surface signatures of the large anticyclone were clearly seen only for about 10 days from 14 September until 24 September 2017, when it became fully ice-filled and indiscernible from the main MIZ region.

The SAR data also reveals the development of several elongated filaments and smaller-scale cyclones on the periphery of the large anticyclone. The latter is frequently reported in literature (e.g. Zatsepin et al., 2019) and is attributed to horizontal shear instabilities of anticyclonic flows that are very effective in producing submesoscale cyclones (McWilliams, 2016). The analysis of the reconstructed current velocity and the filament lengthscale clearly shows submesoscale nature of these features with relative vorticity being about three times the Coriolis frequency. Similarly, strong filaments were recently reported by von Appen et al. (2018) based on detailed field observations in the central FS, showing that in regions where

AW and PW meet such filaments could be deep reaching with substantial vertical motions and density anomaly extending down to 400 m depth, potentially influencing biomass and nutrient distributions in the water column. Such submesoscale processes, as well as wind-induced upwelling in the MIZ, are often associated with intense vertical motions that, in turn, might influence the evolution of the sea ice patterns observed in the SAR data. However, as the time gap between sequential SAR images is < 1 hour, the effect of vertical motions with typical velocities around 0.001-0.01 $\mathrm{m\,s^{-1}}$ (e.g. von Appen et al., 2018) will be negligibly small at such time scales.

As revealed from the SAR data, these elongated filaments move at the maximum speed of 0.75 $\mathrm{m\,s^{-1}}$, being a part of the larger eddy-induced frontogenesis pattern. They are stretching and moving in opposite directions, delineating a dipole eddy structure with opposite vorticity sign. Due to their large spatial extent in the along-front direction, such peculiarities of the submesoscale flow could be hardly resolved even in specialized high-resolution field observations because of their limited spatial coverage. The latter clearly emphasizes the advantage of high-resolution sequential SAR data in resolving small-scale MIZ processes. Given the abundance of MIZ eddies and fronts in the Arctic and Southern Oceans (Kozlov et al., 2019; Swart et al., 2020; von Appen et al., 2018), these energetic features may importantly enhance the vertical heat transport toward the sea ice and influence sea ice melt, upper ocean stratification, and distribution of nutrients and buoyant materials in the water column.

Apart from its potential use in validation and improvement of sea ice forecasting models, the availability of daily, regular, and high-resolution sequential Sentinel-1 SAR observations could contribute to guiding field campaigns and advancing our understanding of multi-scale atmosphere-ice-ocean interactions in MIZs, identifying hot-spots of high kinetic energy, and quantifying lateral and vertical dispersion of various buoyant materials, including microplastics and oil pollution in the polar oceans.

**Data availability.** Sentinel-1 SAR data used in this study can be freely accessed from Copernicus Open Access Hub at https://scihub.copernicus.eu.

**Author contributions.** IEK conceived the research, designed the experiments, performed the data pre-processing and drafted the manuscript. EVP performed the data analysis using the MCC method. IEK, EVP and GEM performed the interpretation of the results, critically revised the text and approved the final version for publication.

**Competing interests.** The authors declare that they have no conflict of interest.

**Acknowledgments.** We are grateful to two anonymous reviewers for their comments and valuable input that improved the paper.

**Financial support.** This study was supported by Russian Science Foundation grant No. 18-77-00082. Software development
for data analysis was partly made under the Ministry of Science and Higher Education of the Russian Federation contract no. 0827-2020-0002. G.E.M. acknowledges support from the United States National Science Foundation, Grant No. 1829969.

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

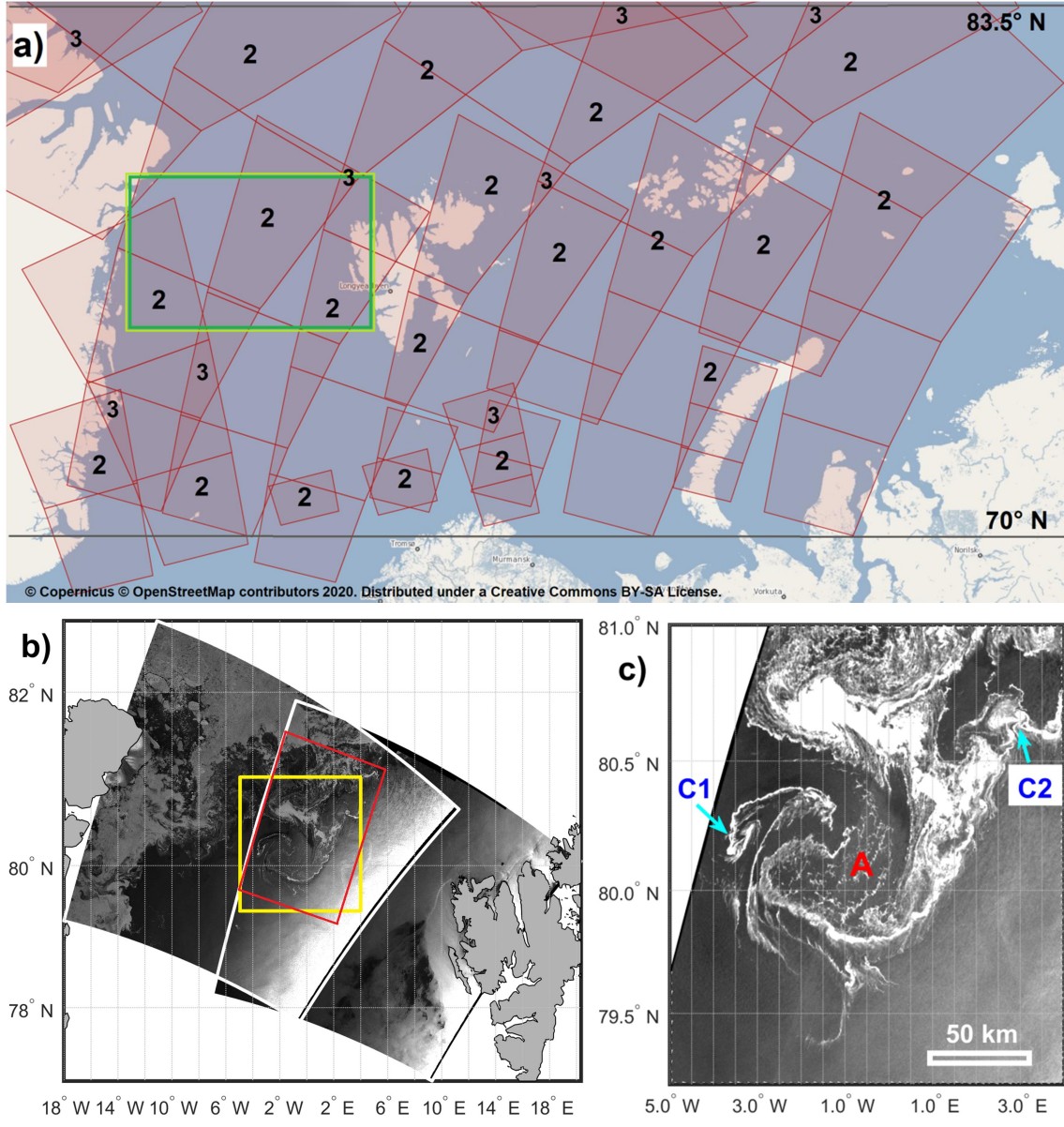

**Figure 1: a) Map of the Europian Arctic showing the coverage of Sentinel-1 A/B SAR image acquisitions available on 17 September 2017. Pink frames mark the borders of individual SAR images, while digits show the number of overlapping SAR frames. Grey lines mark latitude boundaries of 70° N and 83.5° N. Green box shows the area of Fram Strait. The map is taken**
**from Copernicus Open Access Hub © Copernicus © OpenStreetMap 2020. Distributed under a Creative Commons BY-SA License. b) Position of two sequential Sentinel-1 A/B images acquired on September 17, 2017 over Fram Strait. Red and yellow frames mark the regions enlarged in c) and in Figure 2 (a). c) Enlarged fragment of Sentinel-1A image for the same date (07:12 UTC) with distinct signatures of a large anticyclone and several cyclones in the marginal ice zone. Letters A, C1, C2 mark eddies described in the text.**

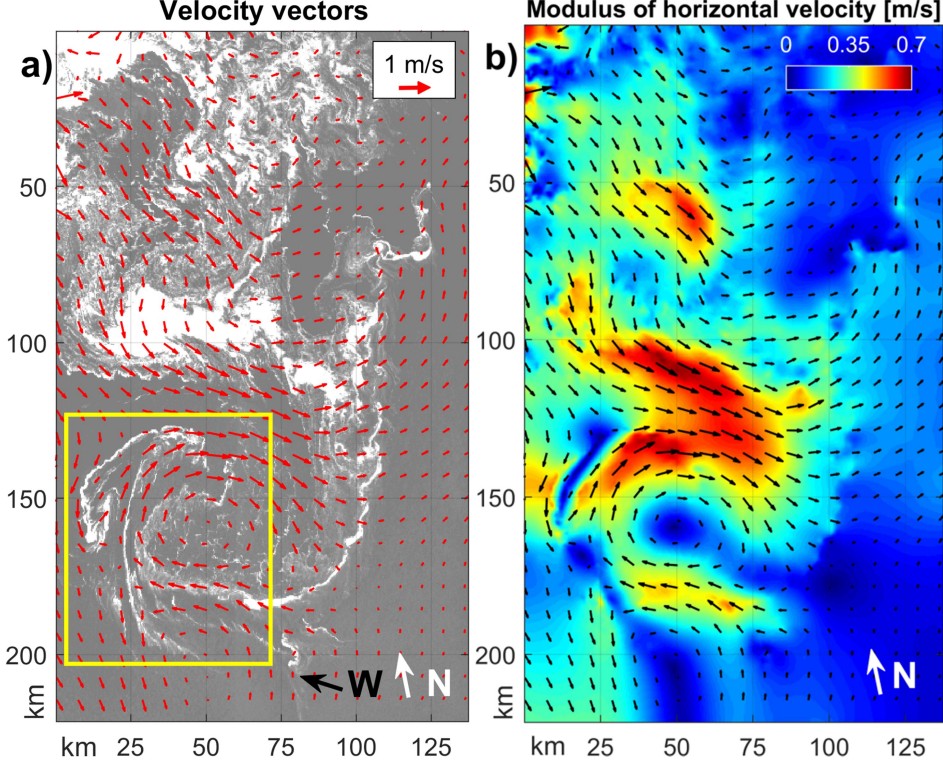

**Figure 2: The horizontal velocity vectors calculated from two sequential Sentinel-1 A/B images acquired on September 17, 2017 over Fram Strait using the MCC method superimposed on a) the SAR image and b) map of the horizontal velocity (speed in color). Letters W and N mark the wind and the northern directions. Yellow frame in (a) shows the region enlarged in Figure 3.**

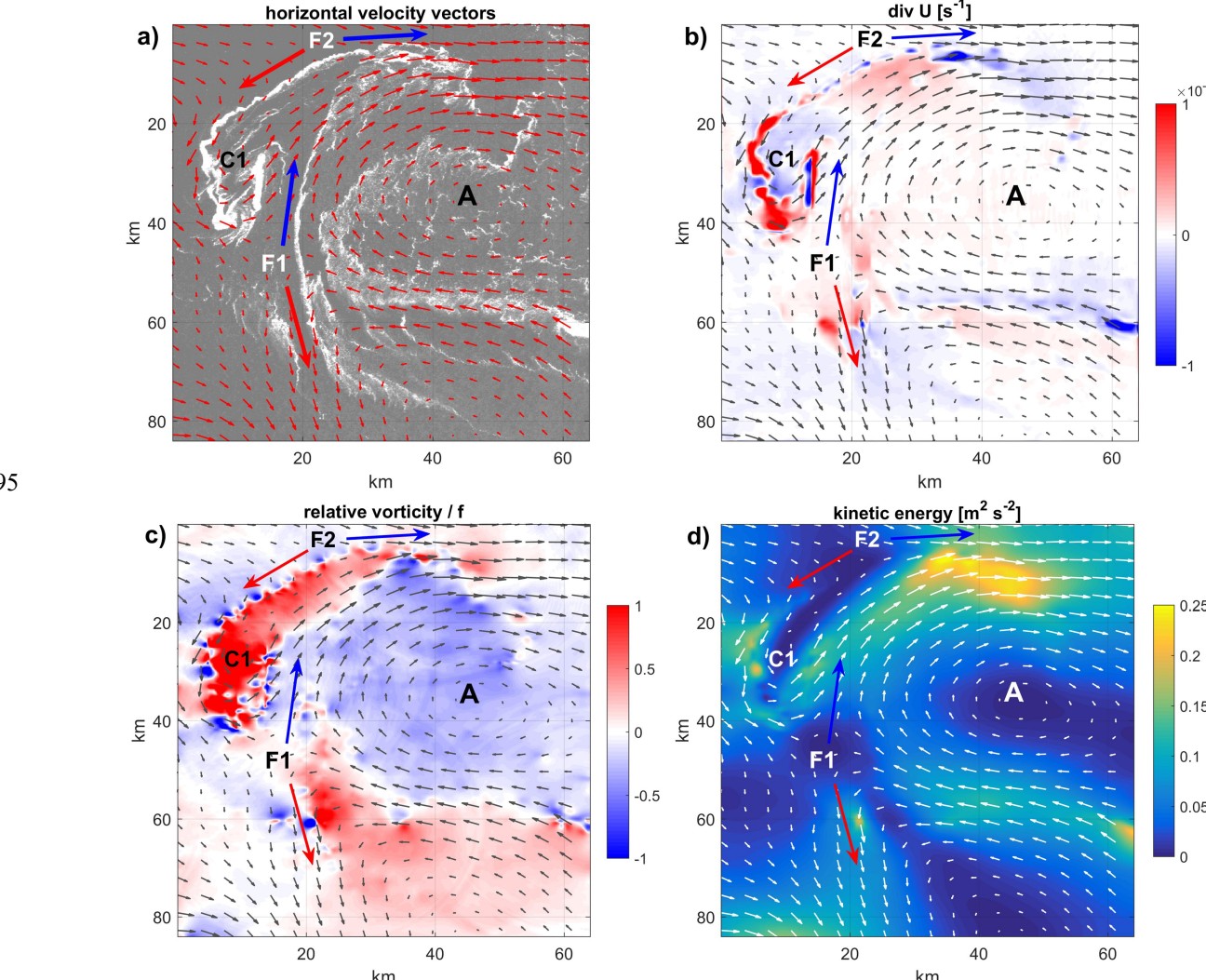

**Figure 3: Results for zoomed area over anticyclone A and cyclone C1: (a) horizontal velocity vectors; (b) horizontal divergence, (c) relative vorticity normalized by the Coriolis parameter; (d) kinetic energy. The largest vector in (a) has a magnitude of 0.75 m s⁻¹. Letters A, C1, F1 and F2 mark the locations of the eddies and filaments described in the text, while blue and red arrows show the movement direction of the filaments F1 and F2.**