# Peer review of "Brief Communication: Mesoscale and submesoscale dynamics in the marginal ice zone from sequential SAR observations"

_The Cryosphere, 2020_

## Referee Comment (RC1) · Wilken-Jon von Appen (Referee) · 11 Jun 2020

The manuscript reports on the use of consecutive (separated by 48 minutes) SAR images of the sea ice distribution in the marginal ice zone of Fram Strait. A method is described to extract the ice velocity at a resolution of hundreds of meters or several cm/s. In the described low ice concentration regions, the ice velocities likely correspond to the ocean velocities underneath. The authors demonstrate that this method can be used to obtain mesoscale and submesoscale oceanic dynamics at unprecedented resolution. This is of high interest both for sea ice physics and physical oceanography as the dynamics of the MIZ depends on both and can only be understood by considering

both. The brief communication is well-written and succinct. I have no major issues with the manuscript, but would like the authors to clarify a few things. That should be straightforward to do. The minor points below should likewise be easy to address. Therefore I recommend minor revision.

Main points:

1. The manuscript impressively demonstrates how well the method works in this one example. However, it would be good to know how usable this method is in general. It would be great if the authors could provide the answers to the following questions in an additional paragraph.

How much effort is it to obtain the velocity vectors for an individual image pair? If it is a lot of work: Can you share the software code such that researchers can run it for their own individual time/location of interest? If it is not much work: Can you make this method operational and provide to others the velocity vectors at all times/locations where appropriate image pairs from Sentinel 1A/B exist?

When can this method be used? What is the range of sea ice concentrations where it applies? Are there differences between seasons in the detail/precision/ease with which the method can be used? E.g. maybe in July (melt season) there is less texture on the sea ice that the satellite could pick up than in September (start of refreezing). Are there influences of weather on the method (e.g. clouds, fog)?

2. The example presented here is from September 2017 in the marginal ice zone in Fram Strait. My high resolution shipboard in-situ study of a submesoscale filament (von Appen et al GRL 2018) was from July 2017, i.e. 3 months earlier. Is there a reason you chose the later time? A direct comparison between the in-situ and the remote sensed data could benefit both methods and reveal more information on the ocean than to consider them separately. I'm not suggesting to change the example presented here, but it might be nice to follow up by also using the method on the July 2017 example, hence also the motivation for the questions under point 1. above.
3. The grammar in the manuscript needs careful editing. Especially articles (a/the) are often missing. I point out a few (but by no means all) of these instances below. I'm not sure whether this should be done now or will take place anyways after acceptance by the journal's copy editors.

Minor points:

l1 title consider "dynamics in the marginal"

l7 "New possibilities for . . . over the marginal . . . are demonstrated"

l8 "within 70-85°N or 70-85°S"

l14 "oceans has been rapidly"

l23 Can that melt rate also be expressed as m/day in the vertical?

l39 Can you give a number what "relatively low concentrations" means (see main point 1 above)?

l45 "independent of"

l51 "has recently become"

l88 "the velocity detection threshold in this case would be 0.03 m s-1" I think it is not just the threshold, but also the precision of your method. I.e. you can only determine the velocity to be 0.03m/s, 0.06m/s, 0.09m/s, and so on. Or am I misunderstanding this?

l93 Did you mean 1150km^2? Otherwise the area would only be 2km long (multiplied by 60km width).

l105 "meaning" How does the second statement (reflects underlying circulation) follow from the first statement (3-5m/s)? Maybe you should state that the winds were very weak or something like that.

l126 ". . . von Appen et al (2018) where velocities of +-0.5m/s were observed with a

vessel mounted acoustic Doppler current profiler."

l130 "in opposite directions"

l148 "one sees very"

l162 "0.02 mˆ2 sˆ-2", i.e. same units as on l149

l165 "instability, of"

l173 "flows which are very effective at producing"

Fig1b Mark box for Fig1c

Fig1c Mark box for Fig2. Otherwise Fig2 would not be georeferenced.

Fig2 Mark box for Fig3. Also consider adding a vector showing the wind direction and a scale vector for 1m/s ocean velocity.

Fig3 Consider to also show strain in a subplot. Also add the "A, C1, F1" letters and the F1 arrows to all subplots to make a comparison easier.

---

## Referee Comment (RC2) · Johnny A. Johannessen (Referee) · 15 Jun 2020

Overall quality:

The paper brings to the attention an interesting ability to examine the mesoscale to submesoscale dynamics in the marginal ice zone using the MCC method on a number of Sentinel-1 A/B SAR acquisitions at short revisit times.

Hitherto the MCC has been frequently applied to optical- and altimeter-based satellite data for studies of mesoscale dynamics in coastal regions. In this paper, the novelty relates to the use of the MCC method to a series of Sentinel-1 SAR image acquisitions

in the marginal ice zone, demonstrating promising results.

The paper is fairly well structured and written, with the inclusion of very good figures.

Scientific issues:

The paper title should preferably be modified to read: Mesoscale and submesoscale dynamics in the marginal ice zone from sequential SAR observations.

The abstract is perhaps too brief. The importance for model validation should be mentioned in consistence with the Discussion and conclusion section.

The submesoscale dynamics are also recognized to have intense, narrow bands of vertical motion. The authors need to address this issue in regard to the application of the MCC method whereby only the estimation of horizontal motion is discussed. For instance, could patterns evolve as influenced by the vertical motion, rather than the horizontal motion. The marginal ice zone is periodically also known to have bands of strong wind induced upwelling that would also influence the subsequent dynamics.

Moreover, the data are collected in September. This is related to the time of year of minimum sea ice extent and concentration. The summer sea ice melt is also nearing an end. Does this set up a shallow mixed layer regime in the MIZ that favors the presence of these mesoscale to submesoscale structures? If so, is there a seasonal variability to these SAR image expressions? It could be good to have this commented and/or addressed.

When Sentinel-1 is mentioned for the first time be more precise; e.g. Sentinel-1 is the European Radar Observatory for the Copernicus joint initiative of the European Commission (EC) and the European Space Agency (ESA).

Technical/editorial corrections:

Line 7: Abstract: …...retrieval in the …….. Line 8: remove…...made…… Line 9: replace….obtaining …with ….estimation of…. Line 11: ….strong sea ice concentration and vorticity... Line 14: ....polar ocean has rapidly grown during.... Line 16: ...major source of uncertainties in the.... Line 22: ...remove......cyclonic.... Line 23:.........ice edge melt rate of.... Line 29:.....large vertical velocities that can entrain.... Line 51: Be consistent in the use of naming convention. Sentinel-1, rather than S-1, is my preference. The former is used in all Figure captions, while it is a mix in the text (although line 51 say...hereafter S-1.... Line 62: .....remove.....from..... Line 63/64: daily basis over region of particular interest, such as the European Arctic Ocean (Fig. 1.a) with 43 S-1 A/B acquisitions available.... Line 66: ...by overlapping SAR scenes. Line 67: ..SAR data , O(100m), ensure a unique.... Line 68: To demonstrate the potential we analyze.... Line 69:....the warm Atlantic Water (AW)..... Line 76: ....SAR images has several steps:... Line 94: and display a large number..... Line 101: ...with model results reported by..... Line 103: .......for surface current estimation using the MCC method. Line 113:.....evolves into the large anticyclonic... Line 128/129: This sentence should be improved. Avoid expressions like.....its movement direction..... Line 146/147: ....for eddies A (ïAÿ≈0.07 f) and C (ïAÿ≈0.3 f) yields a larger Rossby number than for.... Line 163: ...use same unit for EKE in text and Figure Maybe also color scale in the figure could be extended to identify values of 0.3 m2/s2. Line 164:....EKE values were attributed to the complex... Line 173: ....of anticyclonic flows that are very effective...... Line 174: ... use....... relative vorticity ........instead of ......vertical vorticity..... Line 184: .......data to resolve small-scale processes of the complex.... Line 186: ......features may importantly enhance the vertical..... Line 187: ice and influence sea ice melt, upper ocean stratification,...

Figure captions:

Line 254: ....showing the coverage of Sentinel-1 A/B SAR image acquisitions... Line 261 (Figure 2): ....Sentinel-1 A/B images acquired on ... Line 262: .....the SAR image and b) map of the horizontal velocity (speed in color) Line 266: relative vorticity normalized by the Coriolis

Please also note the supplement to this comment:
https://www.the-cryosphere-discuss.net/tc-2020-126/tc-2020-126-RC2-supplement.pdf
* * *

---

## Author Comment (AC1) · 6 Jul 2020

We would like to thank the Reviewer for his positive evaluation of the paper, constructive comments and careful paper reading. Technical issues were addressed directly in the revised version of the manuscript. Below are our answers to the main points raised by the Reviewer:

"How much effort is it to obtain the velocity vectors for an individual image pair? If it is a lot of work: Can you share the software code such that researchers can run it for their own individual time/location of interest? If it is not much work: Can you make this method operational and provide to others the velocity vectors at all times/locations

where appropriate image pairs from Sentinel 1A/B exist?"

In general, our data processing scheme includes two separate tasks – data pre-processing and velocity calculation. The first step includes cutting out the overlapping SAR image fragments, their calibration and normalization, and is planned to be automated soon. The second step (velocity calculation) also needs some additional work with image fragments prior to velocity calculations (e.g. data filtering) and during post-processing (e.g. elimination of false correlations), and currently is done in supervised manner. These two steps are made separately and not yet combined into a single code/procedure. We are working now toward the automatization of the entire procedure to provide velocity vectors to all interested users at least over a single pre-selected site (e.g. Fram Strait), as suggested by the reviewer. Yet, there are still many small issues of the processing chain that are subject of constant improvement, but we hope to finish it asap, at least in the simplified way using the MCC method at the core.

"When can this method be used? What is the range of sea ice concentrations where it applies? Are there differences between seasons in the detail/precision/ease with which the method can be used? E.g. maybe in July (melt season) there is less texture on the sea ice that the satellite could pick up than in September (start of refreezing)? Are there influences of weather on the method (e.g. clouds, fog)?"

We haven't yet tested the method over a very large number of paired SAR images spanning different seasons, background ice concentrations and other environmental factors (like near-surface winds), but our experience from various dates in summer season of 2017 suggests that the MCC works rather effectively for typical sea ice concentrations encountered in the marginal ice zone (20-80%), provided the movement of ice floes is apparent in the sequential SAR images. Though our current experience is lacking to address the question regarding the season, we already plan a more detailed study spanning a longer period of SAR observations over the Fram Strait MIZ. The major issue arising during the processing is the change in the SAR viewing geometry between two sequential scenes. Usually, the desired region of interest would be seen

in the near-range in one image, and in the far-range in another. As a result, the level of radar backscatter (signal strength) over the particular surface area would be different, and so would affect the clarity of eddy manifestation in each of the SAR images in the pair. In such case, individual ice floes shaping the eddy structure might be well seen in one image and poorly seen or have inverted radar contrast in the another one. This issue is addressed during the normalization step, but might be difficult to overcome for very thin ice (either in the beginning of melt season or during freeze onset) whose radar backscatter might become inverted due to the differences in the viewing geometry or varying winds. In regard to weather conditions, SAR is not sensitive to fog and clouds as microwaves effectively penetrate through the atmosphere. Yet, the locally varying near-surface winds may cause some difficulties in the data analysis as described above. If the near-surface winds change over the region of interest during existing time gap between sequential observations, this again might change the radar contrast (signal level) of the sea ice features traced in MCC, resulting e.g. in low correlations and/or inability to retrieve horizontal currents over such ice-covered pixels. Condensed answer to the above two questions is now introduced in the paper.

"The example presented here is from September 2017 in the marginal ice zone in Fram Strait. My high resolution shipboard in-situ study of a submesoscale filament (von Appen et al GRL 2018) was from July 2017, i.e. 3 months earlier. Is there a reason you chose the later time? A direct comparison between the in-situ and the remote sensed data could benefit both methods and reveal more information on the ocean than to consider them separately. I'm not suggesting to change the example presented here, but it might be nice to follow up by also using the method on the July 2017 example, hence also the motivation for the questions under point 1 above."

The only reason here was to show the applicability of the method to retrieve both meso- and submesoscale dynamics in the MIZ. That is why we have chosen specifically the data from 17 September, when the development of the large anticyclone was observed in the MIZ, out of many other paired images in September 2017 (or other months).

[Figure]

We completely agree with the reviewer and would be happy to make a follow-on study considering the entire summer season including the dates when the high-resolution cruise measurements were made in July 2017.

"The grammar in the manuscript needs careful editing. Especially articles (a/the) are often missing. I point out a few (but by no means all) of these instances below. I'm not sure whether this should be done now or will take place anyways after acceptance by the journal's copy editors."

Thank you for pointing the grammar issues of the text, we did our best to improve it.

l23 Can that melt rate also be expressed as m/day in the vertical?

Here we simply cite the facts that are given in the original paper by Johannessen et al. (1987) in the form that emphasizes the horizontal melt rates. As we do not address vertical melt rates further in the paper, the present form seems to be acceptable. However, if the Reviewer insists we can make that change.

l39 Can you give a number what "relatively low concentrations" means (see main point 1 above)? In this part of text we cite the paper by Manucharyan and Thompson (2017), where sea ice concentrations considered were from 50% down to zero.

l88 "the velocity detection threshold in this case would be 0.03 m s-1" I think it is not just the threshold, but also the precision of your method. I.e. you can only determine the velocity to be 0.03m/s, 0.06m/s, 0.09m/s, and so on. Or am I misunderstanding this?

In fact, this '0.03 m s-1' is the lower limit below which we can't resolve the object's movement working with S-1 GRD EW mode images. The precision of velocity calculations is then set up by the pixel spacing (equal to 40 m) which equals to 40 m/48 min = $\pm$ 0.01 m s-1. This is now added to the text.

l93 Did you mean 1150kmȨ̈Ȩ2? Otherwise the area would only be 2km long (multiplied by 60 km width).

Thank you for noting this typo, the correct value is $\sim$ 15 000 km^2 as the average length of the MIZ was about 220 km long (being 60 km iwde).

"meaning" How does the second statement (reflects underlying circulation) follow from the first statement (3-5m/s)? Maybe you should state that the winds were very weak or something like that.

Thank you, we have slightly rephrased this sentence. Now it sounds: "According to WindSat and ASCAT scatterometers' data for 17 September 2017 (not shown), the wind conditions were characterized by low south-easterly winds of 3-5 m s-1 under which the ice drift near the ice edge should reflect the underlying ocean circulation (Shuchman et al., 1987; Manucharyan and Thompson, 2017)."

l165 "instability, of"

Here we meant "the barotropic and baroclinic instability of an ice edge jet...". Shall it be separated by comma in this case?

Fig3 Consider to also show strain in a subplot. Also add the "A, C1, F1" letters and the F1 arrows to all subplots to make a comparison easier.

Thank you for the suggestion. We have incorporated all the suggested changes into the figures. Yet, we are already at the limit of the word count and paper length for the manuscript type "Brief Communication". We, therefore, have no extra space to accommodate one more subplot and its description in the text. This can only be done at the expense of other material in the paper.
* * *
[Figure]

b)

**Fig. 1.** Updated Fig.1b

[Figure]

**Fig. 2.** Updated Fig.2a

[Figure]

**Fig. 3.** Updated Fig.3

---

## Author Comment (AC2) · 7 Jul 2020

We would like to thank the Reviewer for his positive evaluation of the paper, constructive comments and helping us to improve the quality of the text by specifying many text edits that were fixed directly in the revised version of the manuscript. Below are our answers to the main points raised by the Reviewer:

"The paper title should preferably be modified to read: Mesoscale and submesoscale dynamics in the marginal ice zone from sequential SAR observations. The abstract is perhaps too brief. The importance for model validation should be mentioned in consistence with the Discussion and conclusion section."

[Figure]

Thank you for the comments. According to the Cryosphere journal rules, all short papers must be entitled starting with "Brief Communication: ...". We completely agree with changing of 'of the MIZ' to 'in the MIZ'. Regarding the abstract length, it was made according to the journal rules for such paper types when the abstract is limited by 100 words only. These rules are specified at the journal homepage at https://www.the-cryosphere.net/about/manuscript_types.html. Nevertheless, we have managed to mention the 'model validation' in the abstract which now counts for 97 words.

"The submesoscale dynamics are also recognized to have intense, narrow bands of vertical motion. The authors need to address this issue in regard to the application of the MCC method whereby only the estimation of horizontal motion is discussed. For instance, could patterns evolve as influenced by the vertical motion, rather than the horizontal motion. The marginal ice zone is periodically also known to have bands of strong wind induced upwelling that would also influence the subsequent dynamics."

Thank you for this important point. We agree, intense vertical motions associated with submesoscale dynamics may impact the evolution of sea ice patterns observed in the SAR images. This might either be due to upwelling of relatively warm subsurface water to the ocean-ice interface in the narrow surface current divergence zones resulting in gradual sea ice melt at the surface, or the surface water subduction in the current convergence zones that was well described e.g. by von Appen et al. (2018) in the study region from in situ sampling. In fact, this is something we have briefly mentioned in the introduction and addressed in more details when showing the horizontal divergence field in Fig. 3b. The latter clearly shows the formation of intense surface current convergence/divergence zones where sea ice accumulates/repel and the surface water goes down/up. In this sense, surface current retrieval using the MCC not only shows the regions of intense horizontal currents, but also the surface areas with intense vertical motions marked by high surface current divergence/convergence values. On the another hand, we might expect that the time interval between sequential

SAR acquisitions (around 1 hour) is perhaps too short to see the direct impact of the vertical motions on the evolution of sea ice signatures seen in the data. Do we correctly understand this comment or have missed something? Part of this answer is now included to the revised paper version.

"Moreover, the data are collected in September. This is related to the time of year of minimum sea ice extent and concentration. The summer sea ice melt is also nearing an end. Does this set up a shallow mixed layer regime in the MIZ that favors the presence of these mesoscale to submesoscale structures? If so, is there a seasonal variability to these SAR image expressions? It could be good to have this commented and/or addressed."

Thanks for pointing this. Yes, we agree that our observations made in mid-September might have some specific seasonal features, like the mentioned formation of the shallow mixed layer favorable for the generation of such dynamic features in the MIZ. Yes, we do expect that there is a certain seasonal variability of such SAR image expressions. This is confirmed, for example, by results of our recent SAR-based study in the Western Arctic Ocean, where analysis of the data from June to October showed a clear peak in the number of MIZ eddies in September-October of 2007, 2011 and 2016. Moreover, Bondevik (2011) in her Master thesis also showed that SAR-based detections of MIZ eddies along the East Greenland Current throughout the years of 2008 and 2009 had a distinct seasonal variability with highest eddy numbers observed from May to September. In fact, we plan to address this question in more details for the Fram Strait MIZ in future by considering a longer time period of SAR observations. Some of the points mentioned above will be added to the revised paper version. We had a reference to Bondevik (2011) work in our initial paper version, but due to the limit of 20 references we had to exclude it. If the Reviewer feels it is necessary to have it, we will try to insert it.

"When Sentinel-1 is mentioned for the first time be more precise; e.g. Sentinel-1 is the European Radar Observatory for the Copernicus joint initiative of the European

[Figure]

Commission (EC) and the European Space Agency (ESA)."

Thank you for the comment. We had no space to accommodate for this info in the abstract, yet, have given the full description in the main text when Sentinel-1 was first mentioned.

"Line 128/129: This sentence should be improved. Avoid expressions like. . .its movement direction. . ."

We have changed this to ". . .its propagation direction. . .", however, we don't know if this has improved the overall sounding of the sentence.

"Line 163: . . .use same unit for EKE in text and Figure Maybe also color scale in the figure could be extended to identify values of 0.3 m2/s2".

Sorry, we couldn't get this as we have the same kinetic energy units expressed in the text and in the figure given in m2 s-2. Following your advice, we have adjusted the color scale to 0.25 m2/s2 , as this gives a better visual expression as compared to the initial 0.25 m2/s2 and suggested 0.3 m2/s2. We had also to add A, C1, F1, F2 notions and arrows to all subplots of Fig.3 according to the another reviewer suggestion.

[Figure]

[Figure]

[Figure]

**Fig. 1.**

---

## Editor Decision (ED1)

The manuscript reports on the use of consecutive (separated by 48 minutes) SAR images of the sea ice distribution in the marginal ice zone of Fram Strait. A method is described to extract the ice velocity at a resolution of hundreds of meters or several cm/s. In the described low ice concentration regions, the ice velocities likely correspond to the ocean velocities underneath. The authors demonstrate that this method can be used to obtain mesoscale and submesoscale oceanic dynamics at unprecedented resolution. This is of high interest both for sea ice physics and physical oceanography as the dynamics of the MIZ depends on both and can only be understood by considering

both. The brief communication is well-written and succinct. I have no major issues with the manuscript, but would like the authors to clarify a few things. That should be straightforward to do. The minor points below should likewise be easy to address. Therefore I recommend minor revision.

Main points:

1. The manuscript impressively demonstrates how well the method works in this one example. However, it would be good to know how usable this method is in general. It would be great if the authors could provide the answers to the following questions in an additional paragraph.

How much effort is it to obtain the velocity vectors for an individual image pair? If it is a lot of work: Can you share the software code such that researchers can run it for their own individual time/location of interest? If it is not much work: Can you make this method operational and provide to others the velocity vectors at all times/locations where appropriate image pairs from Sentinel 1A/B exist?

When can this method be used? What is the range of sea ice concentrations where it applies? Are there differences between seasons in the detail/precision/ease with which the method can be used? E.g. maybe in July (melt season) there is less texture on the sea ice that the satellite could pick up than in September (start of refreezing). Are there influences of weather on the method (e.g. clouds, fog)?

2. The example presented here is from September 2017 in the marginal ice zone in Fram Strait. My high resolution shipboard in-situ study of a submesoscale filament (von Appen et al GRL 2018) was from July 2017, i.e. 3 months earlier. Is there a reason you chose the later time? A direct comparison between the in-situ and the remote sensed data could benefit both methods and reveal more information on the ocean than to consider them separately. I'm not suggesting to change the example presented here, but it might be nice to follow up by also using the method on the July 2017 example, hence also the motivation for the questions under point 1. above.

3. The grammar in the manuscript needs careful editing. Especially articles (a/the) are often missing. I point out a few (but by no means all) of these instances below. I'm not sure whether this should be done now or will take place anyways after acceptance by the journal's copy editors.

Minor points:

l1 title consider "dynamics in the marginal"

l7 "New possibilities for . . . over the marginal . . . are demonstrated"

l8 "within 70-85°N or 70-85°S"

l14 "oceans has been rapidly"

l23 Can that melt rate also be expressed as m/day in the vertical?

l39 Can you give a number what "relatively low concentrations" means (see main point 1 above)?

l45 "independent of"

l51 "has recently become"

l88 "the velocity detection threshold in this case would be 0.03 m s-1" I think it is not just the threshold, but also the precision of your method. I.e. you can only determine the velocity to be 0.03m/s, 0.06m/s, 0.09m/s, and so on. Or am I misunderstanding this?

l93 Did you mean 1150km^2? Otherwise the area would only be 2km long (multiplied by 60km width).

l105 "meaning" How does the second statement (reflects underlying circulation) follow from the first statement (3-5m/s)? Maybe you should state that the winds were very weak or something like that.

l126 ". . . von Appen et al (2018) where velocities of +-0.5m/s were observed with a

vessel mounted acoustic Doppler current profiler."

l130 "in opposite directions"

l148 "one sees very"

l162 "0.02 m^2 s^-2", i.e. same units as on l149

l165 "instability, of"

l173 "flows which are very effective at producing"

Fig1b Mark box for Fig1c

Fig1c Mark box for Fig2. Otherwise Fig2 would not be georeferenced.

Fig2 Mark box for Fig3. Also consider adding a vector showing the wind direction and a scale vector for 1m/s ocean velocity.

Fig3 Consider to also show strain in a subplot. Also add the "A, C1, F1" letters and the F1 arrows to all subplots to make a comparison easier.
* * *
**Interactive comments on the paper "Mesoscale and submesoscale dynamics of marginal ice zone from sequential SAR observations" submitted by Igor E. Kozlov et al, 2020.**

**Overall quality:**

The paper brings to the attention an interesting ability to examine the mesoscale to submesoscale dynamics in the marginal ice zone using the MCC method on a number of Sentinel-1 A/B SAR acquisitions at short revisit times.

Hitherto the MCC has been frequently applied to optical- and altimeter-based satellite data for studies of mesoscale dynamics in coastal regions. In this paper, the novelty relates to the use of the MCC method to a series of Sentinel-1 SAR image acquisitions in the marginal ice zone, demonstrating promising results.

The paper is fairly well structured and written, with the inclusion of very good figures.

**Scientific issues:**

The paper title should preferably be modified to read: Mesoscale and submesoscale dynamics in the marginal ice zone from sequential SAR observations.

The abstract is perhaps too brief. The importance for model validation should be mentioned in consistence with the Discussion and conclusion section.

The submesoscale dynamics are also recognized to have intense, narrow bands of vertical motion. The authors need to address this issue in regard to the application of the MCC method whereby only the estimation of horizontal motion is discussed. For instance, could patterns evolve as influenced by the vertical motion, rather than the horizontal motion. The marginal ice zone is periodically also known to have bands of strong wind induced upwelling that would also influence the subsequent dynamics.

Moreover, the data are collected in September. This is related to the time of year of minimum sea ice extent and concentration. The summer sea ice melt is also nearing an end. Does this set up a shallow mixed layer regime in the MIZ that favors the presence of these mesoscale to submesoscale structures? If so, is there a seasonal variability to these SAR image expressions? It could be good to have this commented and/or addressed.

When Sentinel-1 is mentioned for the first time be more precise; e.g. Sentinel-1 is the European Radar Observatory for the Copernicus joint initiative of the European Commission (EC) and the European Space Agency (ESA).

**Technical/editorial corrections:**

Line 7: Abstract: …..retrieval in the …….
Line 8: remove…..made……
Line 9: replace….obtaining …with ….estimation of….
Line 11: ….strong sea ice concentration and vorticity…
Line 14: ….polar ocean has rapidly grown during….
Line 16: …major source of uncertainties in the….

Line 22: ….remove……cyclonic….
Line 23:………ice edge melt rate of….
Line 29:…..large vertical velocities that can entrain….
Line 51: Be consistent in the use of naming convention. Sentinel-1, rather than S-1, is my preference. The former is used in all Figure captions, while it is a mix in the text (although line 51 say…hereafter S-1….
Line 62: …..remove…..from…..
Line 63/64: daily basis over region of particular interest, such as the European Arctic Ocean (Fig. 1.a) with 43 S-1 A/B acquisitions available….
Line 66: …by overlapping SAR scenes.
Line 67: ..SAR data , O(100m), ensure a unique….
Line 68: To demonstrate the potential we analyze….
Line 69:….the warm Atlantic Water (AW)…..
Line 76: ….SAR images has several steps:…
Line 94: and display a large number…..
Line 101: …with model results reported by…..
Line 103: …….for surface current estimation using the MCC method.
Line 113:…..evolves into the large anticyclonic…
Line 128/129: This sentence should be improved. Avoid expressions like…..its movement direction…..
Line 146/147: ….for eddies A ($\xi \approx 0.07$ f) and C ($\xi \approx 0.3$ f) yields a larger Rossby number than for….
Line 163: …use same unit for EKE in text and Figure Maybe also color scale in the figure could be extended to identify values of 0.3 $m^2/s^2$.
Line 164:….EKE values were attributed to the complex…
Line 173: ….of anticyclonic flows that are very effective……
Line 174: … use……. relative vorticity ……..instead of ……vertical vorticity…..
Line 184: …….data to resolve small-scale processes of the complex….
Line 186: ……features may importantly enhance the vertical…..
Line 187: ice and influence sea ice melt, upper ocean stratification,…

Figure captions:

Line 254: ….showing the coverage of Sentinel-1 A/B SAR image acquisitions…
Line 261 (Figure 2): ….Sentinel-1 A/B images acquired on …
Line 262: …..the SAR image and b) map of the horizontal velocity (speed in color)
Line 266: relative vorticity normalized by the Coriolis